# Inherited/Genetically-Associated Pheochromocytoma/ Paraganglioma Syndromes and COVID-19

**DOI:** 10.3390/medicina57101033

**Published:** 2021-09-28

**Authors:** Ioannis Ilias, Gregory Kaltsas, Konstantinos Barkas, George P. Chrousos

**Affiliations:** 1Department of Endocrinology, Diabetes and Metabolism, “Elena Venizelou” Hospital, GR-11521 Athens, Greece; 21st Department of Propaedeutic and Internal Medicine, National and Kapodistrian University of Athens, GR-11527 Athens, Greece; gregory.kaltsas@gmail.com; 3Department of Neurosurgery, “Agios Panteleimon” General Hospital of Nikaia-Pireas, GR-12351 Athens, Greece; kbarkas.neurosurgery@gmail.com; 4University Research Institute of Maternal and Child Health & Precision Medicine, National and Kapodistrian University of Athens, “Aghia Sophia” Children’s Hospital, GR-11527 Athens, Greece; chrousos@gmail.com; 5UNESCO Chair on Adolescent Health Care, National and Kapodistrian University of Athens, “Aghia Sophia” Children’s Hospital, GR-11527 Athens, Greece

**Keywords:** adrenal gland neoplasms, pheochromocytoma, angiotensin II, hypoxia-inducible factor 1, alpha subunit, angiotensin-converting enzyme 2, COVID-19

## Abstract

In some subjects with inherited pheochromocytoma/paraganglioma (PPG) syndromes, hypoxia-inducible factor 1 alpha (HIF1α) stabilization/activation could lead to an increase in angiotensin converting enzymes (ACE). This would result in the stimulation of angiotensin (AT) II production and, hence, reduce the availability of ACE 2. The latter would provide decreased numbers of binding sites for the spike protein of SARS-CoV-2 and, therefore, result in less points of viral entry into cells. Thus, subjects with HIF1α-associated PPG syndromes may benefit from an inherent protective effect against COVID-19. Such an implication of HIF1α vis-à-vis COVID-19 could open ways of therapeutic interventions.

The diagnosis and treatment of pheochromocytomas/paragangliomas (PPG) can be challenging [1]; concomitant COVID-19 can render these tasks more difficult [2].

Inherited/genetically associated PPG syndromes, particularly those caused by mutations in the succinate dehydrogenase (SDH) B or D genes or the von Hippel-Lindau (VHL) gene, are associated with the stabilization/activation of the hypoxia-inducible factor 1 alpha (HIF1α). This factor is implicated in promoting angiogenesis and tumour growth [3,4].

High-altitude dwelling with hypoxia may lead to physiological adaptations that include morphological and functional alterations in the carotid bodies, HIF1α stabilization/activation and upregulation of angiotensin-converting enzyme (ACE) expression [5,6,7,8,9,10]. Recent reports indicate that living in high altitudes entails some degree of protection from SARS-CoV-2 infection/COVID-19, possibly by intermittent activation of HIF1α [11,12,13]. The latter also suppresses the expression of angiotensin-converting enzyme 2 (ACE2), which is the receptor that permits the cellular entry of SARS-CoV-2 [14,15,16].

Bearing all the above in mind, in some subjects with inherited PPG syndromes, HIF1α stabilization/activation could lead to an increase in ACE. This would result in stimulation of angiotensin (AT) II production and, hence, reduced availability of ACE 2. The latter would provide decreased numbers of binding sites for the spike protein of SARS-CoV-2 and, therefore, less points of viral entry into cells. Thus, subjects with HIF1α-associated PPG syndromes may benefit from an inherent protective effect against COVID-19. A study of PPG patients that are followed in specialized centres worldwide would be informative; if a protective effect is noted, the implication of HIF1α in COVID-19 could open ways of therapeutic interventions.

In conclusion, persons dwelling in high altitudes, or patients with PPG, may be protected to a degree from COVID-19, by means of HIF1α activation. The latter could be a target of potential COVID-19 therapies.

## Data Availability

Not applicable.

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
