# Peer review of "Inherited/Genetically-Associated Pheochromocytoma/ Paraganglioma Syndromes and COVID-19"

_medicina, 2021, doi:10.3390/medicina57101033_

Round 1
Reviewer 1 Report
In the present commentary ILIAS et al., focus on PPG patients and Covid19 and describe how HIF1α-associated PPG syndromes may benefit from an inherent protective effect against COVID-19. Since this manuscript is a commentary, I do not have any scientific comments.
However, it is very difficult to understand the rationale of this commentary and it will be better if authors could describe the purpose of the commentary. Also, it is not clear which article this commentary comment about?
Author Response
We thank the Reviewer for his/her comments.
We are sorry that (s)he finds it difficult to understand the rationale of our contribution.
The premise delineated in this contribution is as follows:
[1]. Persons dwelling in high altitude may be protected to a degree from Covid-19
[2]. These high-altitude inhabitants are noted for having HIF1alpha activation
[3]. Subjects with some genetically associated PPGs also have HIF1alpha activation
[4]. Thus these subjects with PPG may also be protected from Covid-19
[5]. If the above is true then HIF1alpha may be a target of potential Covid-19 therapies.
The purpose of our commentary is to elaborate on this premise; since the relevant literature is not limitless we feel that such a format is more appropriate than either a Review article (being too long) or a Letter to the editor (being too short). Alternatively, this contribution could be termed a hypothesis, opinion or editorial piece.
This is a standalone manuscript that does not refer to any particular article from the Journal; it was solicited for a Covid-19-themed issue of this Journal.
In the Revised version of the Manuscript, to clarify the paper's rationale, we have added the following in the article's Conclusion:
"In conclusion, persons dwelling in high altitude or patients with PPG may be protected to a degree from Covid-19, by means of HIF1α activation. The latter could be a target of potential Covid-19 therapies."
Reviewer 2 Report
Ilias et al. wrote an interesting commentary on the relationship between PPG syndrome and the SARS COVID-19 infection focusing on the molecular pathway of HIF1 alfa whose stabilization/activation could be protective against COVID-19. Authors' considerations could open the way to further studies on this topic with important implication COVID-19 prevention. Therefore it is suitable for publication.
Author Response
We thank the Reviewer for his/her comments.